# A Proposal for the Classification of Temporomandibular Joint Disc Deformity in Hemifacial Microsomia

**DOI:** 10.3390/bioengineering10050595

**Published:** 2023-05-16

**Authors:** Xiaochen Xue, Zhixu Liu, Hongpu Wei, Xudong Wang

**Affiliations:** 1Department of Oral and Cranio-Maxillofacial Surgery, Shanghai Ninth People’s Hospital, Shanghai Jiao Tong University School of Medicine, Shanghai 200011, China; 2National Clinical Research Center for Oral Disease, Shanghai Key Laboratory of Stomatology, Shanghai Research Institute of Stomatology, Shanghai 200011, China

**Keywords:** hemifacial microsomia, temporomandibular joint disc, OMENS+

## Abstract

Hemifacial microsomia (HFM) is the second most common congenital craniofacial disease and has a wide spectrum of symptoms. The classic diagnostic criterion for hemifacial microsomia is the OMENS system, which was later refined to the OMENS+ system to include more anomalies. We analyzed the data of 103 HFM patients with magnetic resonance imaging (MRI) for temporomandibular joint (TMJ) discs. The TMJ disc classification was defined into four types: D0 for normal disc size and shape; D1 for disc malformation with adequate length to cover the (reconstructed) condyle; D2 for disc malformation with inadequate length to cover the (reconstructed) condyle; and D3 for no obvious presence of a disc. Additionally, this disc classification was positively correlated with the mandible classification (correlation coefficient: 0.614, *p* < 0.01), ear classification (correlation coefficient: 0.242, *p* < 0.05), soft tissue classification (correlation coefficient: 0.291, *p* < 0.01), and facial cleft classification (correlation coefficient: 0.320, *p* < 0.01). In this study, an OMENS+D diagnostic criterion is proposed, confirming the conjecture that the development of the mandibular ramus, ear, soft tissue, and TMJ disc, as homologous and adjacent tissues, is affected to a similar degree in HFM patients.

## 1. Introduction

Hemifacial microsomia (HFM) refers to a series of complicated and varied symptoms caused by malformation of the first and second branchial arches during embryogenesis, with different names, such as craniofacial microsomia, first and second branchial arch syndrome, otomandibular dysostosis, and auriculobranchiogenic dysplasia [1]. It is the second most common facial congenital defect, following cleft lip/palate, with an incidence estimated at 1 in 3000 to 1 in 5600 live births [1,2,3]. Most hemifacial microsomia cases occur sporadically and unilaterally, with bilateral hypoplasia noted in 5–30% of cases [1]. HFM involves hard tissues such as the upper and lower jaws and related soft tissues and can be manifested as a series of malformations. Eye malformations include strabismus, anophthalmia, microphthalmia, eye asymmetry, blepharoptosis, and exophthalmos. Auricular abnormalities include appendage, preauricular fistula, microtia, ear asymmetry, and external auditory canal atresia. Malformations of the first and second pharyngeal arches include cleft lip and palate, cleft tongue, maxillary and mandibular hypoplasia, and malocclusion [4]. Additionally, HFM patients often have extracranial defects, with neurological abnormalities present in 5% to 15% of patients, cardiac abnormalities present in 14% to 47% of patients, urogenital abnormalities present in 5% to 6% of patients, lung and gastrointestinal abnormalities present in 10% of patients, and skeletal abnormalities present in 40% to 60% of patients [5,6,7,8]. Therefore, a comprehensive evaluation of hearing assessment, speech analysis, psychosocial assessment, and spinal imaging should be performed based on the patient’s symptoms [9].

To better describe and classify the wide spectrum of symptoms, several grading systems have been proposed. The Pruzansky [10] and Pruzansky–Kaban [11] classifications focus on mandible defects: Type I: normal morphologic characteristics of the ramus and condyle but diminished in size. Type II: significant architectural and size distortion of the ramus, condyle, and sigmoid notch; Type IIA: acceptable glenoid fossa anatomy and position with respect to the unaffected side; Type IIB: temporomandibular joint (TMJ) malposition. Type III: gross distortion or complete agenesis of the ramus. Meanwhile, the Lauritzen [12] classification evaluates not only the mandible but also the entire facial area with simple standards: Type I: facial skeleton is complete; TMJ is intact and functional; mandible asymmetrical, creating a lack of fullness on one side of the face; Type IA: mild; Type IB: severe (depending on the degree of asymmetry). Type II: the condylar head is missing with no functional TMJ; zygomatic arch present; adequate support of a reconstructed condyle and fossa construction is not necessary. Type III: condyle is missing, and zygomatic arch is hypoplastic or absent; glenoid fossa is absent or rudimentary and must be constructed. Type IV: in addition to Type III defects, the lateral and inferior orbital rims are grossly recessed posteriorly. Type V: in addition to Type IV defects, the orbit is dystopic and frequently hypoplastic; the neurocranium is asymmetrical with a flat temporal fossa. Meurman [13] and Murray et al. [14] studied ear and soft tissue defects, respectively. The SAT classification was introduced based on these previous grading systems [7]. Then, Vento et al. took a further step and proposed what is currently the most agreed upon “OMENS” classification system [15], which was later refined to “OMENS+” to include more anomalies [5].

In “OMENS+”, each letter stands for a kind of defect, as shown in Table 1. “O” stands for orbital asymmetry, evaluating both orbital size and position. “M” stands for mandibular hypoplasia, evaluating the size and shape of the ramus, condyle, and glenoid fossa. “E” stands for external ear deformity. “N” stands for facial nerve dysfunction. “S” stands for soft tissue deficiency. “+” stands for the presence of associated anomalies, and macrostomia/transverse facial cleft is frequently mentioned and written as “+C” [1,5,9,15,16,17].

Studying magnetic resonance imaging (MRI) of the temporomandibular joint (TMJ) is necessary to evaluate the TMJ’s condition and to make treatment plans for HFM patients. Nebbe et al. first superimposed sagittal MRI to lateral cephalometric radiographs to evaluate the TMJ disc position in 1998 [18], and since then, MRI has gradually become the standard for viewing the cartilaginous disc of the TMJ. When analyzing the TMJ MRI of hemifacial microsomia patients, we found TMJ disc malformation, malposition, and even absence in many cases. Anterior disc displacement with or without reduction could be observed in patients with enough TMJ mobility. Additionally, for patients with little or no mobility of the TMJ, discs are usually in the wrong places compared to normal ones because of the malformation of the glenoid fossa and condyle. In our study, we would like to focus on the disc’s length considering the potential condyle–disc relationship with a condyle or reconstructed condyle. An improper condyle–disc relationship may cause facial asymmetry or malocclusion [19], which leads to less stability of surgical and orthodontic treatment for HFM patients.

In this study, we aimed to fulfill the OMENS+ system with a TMJ disc classification and investigate the correlation between disc and OMENS+C classifications. Our goal is to bring attention to the TMJ disc in hemifacial microsomia patients in order to aid in comprehensive surgery planning and prognosis for the stability of surgical treatment and orthodontic treatment.

## 2. Materials and Methods

### 2.1. Study Population

A total of 103 consecutive patients diagnosed with unilateral or bilateral hemifacial microsomia who were referred to the Department of Oral and Craniomaxillofacial Surgery in Shanghai Ninth People’s Hospital were chosen. Medical charts, including age, gender, affected side, family history, photographs, and radiographs—including cephalometric films, panoramic radiographs, computed tomography (CT) with three-dimensional (3D) reconstruction, and TMJ MRI—were reviewed. Patients with incomplete chart documentation of their deformities based on the OMENS+ system were excluded.

Among the 103 patients, there were 46 females and 57 males, aged 5 to 39 years old.

### 2.2. CT, 3D Reconstruction, and MRI

CT scans were obtained using a 1.25 mm thick slice with a hospital-based spiral scanner (GE Healthcare). The CT data were imported into ProPlan CMF 2.0 (Materialise) for 3D reconstruction [20,21].

MRI scans were obtained using a 1.5 T imager (Signa, General Electric, Milwaukee, WI) with bilateral 3-inch TMJ surface coil receivers [22]. The transection plane was scanned to find the long axis of the condyle, and the sagittal plane was then determined to be perpendicular to this long axis [23].

### 2.3. Statistical Analysis

For bilateral patients, we defined both affected mandibles and discs as two single samples during statistical analysis. The correlation between orbit, mandible, external ear, facial nerve, soft tissue, facial cleft, and disc classifications was analyzed with the Spearman rank correlation coefficient test (2-tailed).

## 3. Results

### 3.1. Classification of the OMENS+C

All 103 patients were evaluated and classified according to the OMENS+C system [1]. We present the mandible classification as an example here. M0: normal mandible; M1: small mandible and glenoid fossa with short ramus (Figure 1a); M2a: abnormally shaped and short ramus (glenoid fossa in acceptable position) (Figure 1b); M2b: abnormally shaped and short ramus (glenoid fossa is inferiorly, medially, and anteriorly displaced with severe hypoplasia of condyle) (Figure 1c); M3: absence of ramus and glenoid fossa (Figure 1d). Besides mandibular defects, the orbital, ear, facial nerve, and soft tissue defects and facial cleft were also evaluated.

### 3.2. Definition of Disc Classification

Studies have shown that untreated TMJ disc displacement may lead to an unstable maxilla–mandibular complex after orthognathic surgery and cause skeletal relapse related to condylar remodeling and resorption [24,25]. These studies indicated a possible instability of the mandible after surgical repositioning. Therefore, a proper condyle–disc relationship should be expected not only in orthognathic surgery but also in mandibular distraction osteogenesis (MDO) and costochondral grafting (CCG), which are currently the major surgical methods for HFM. The sagittal slice with the largest cross-section of the TMJ disc was chosen to evaluate the disc classification in each patient on the affected side. The evaluation criteria included the shape, length, and presence of the disc. We did not take the preoperative condyle and disc position into consideration, because most HFM patients have condylar deformities and malpositioned discs. Instead, we focused on the potential condyle–disc relationship, which includes the current or reconstructed condyle and the disc’s length.

We defined the disc classification as follows:

D0: Normal disc size and shape (Figure 2). The TMJ disc is marked with a dotted white line in the following figures;

D1: Disc malformation with adequate length to cover the (reconstructed) condyle (Figure 3). A proper condyle–disc relationship could be expected after a certain treatment in this type;

D2: Disc malformation with inadequate length to cover the (reconstructed) condyle (Figure 4). After a certain treatment, an acceptable condyle–disc relationship could be expected in this type;

D3: No obvious presence of disc (Figure 5). No condyle–disc relationship could be observed in this type. Treatment stability may be influenced.

### 3.3. Sample Distribution

We analyzed the distribution of OMENS+C and disc classifications in 108 samples (Table 2), as we defined both affected sides as two single samples in bilateral HFM patients.

The sample distribution of orbital defects shows that O0 (32.4%) and O2 (30.6%) are the most common types, while O1 (22.2%) and O3 (14.8%) types are less commonly observed. Regarding mandibular defects, M2a (32.4%) and M2b (41.7%) are the most common types, while M1 (13.9%) and M3 (12.0%) samples are fewer in number. Regarding external ear defects, E0 (33.3%), E2 (25.0%), and E3 (30.6%) are all the most common types, while the E1 (11.1%) type is observed less. Regarding facial nerve defects, N0 (77.8%) is the most common type, while N1 (9.3%), N2 (10.2%), and N3 (2.8%) are all observed less. Regarding soft tissue defects, S0 (27.8%), S1 (38.0%), and S2 (28.7%) are all the most common types, while S3 (5.6%) samples are fewer in number. Regarding facial clefts, C0 (72.2%) is the most common type, while C1 (19.4%) and C2 (8.3%) are observed less. Regarding TMJ disc defects, D0 (28.7%), D2 (33.3%), and D3 (22.2%) are the most common types, while D1 (15.7%) samples are fewer in number.

Most of the patients who came to our department were mainly seeking medical advice for malformed or asymmetrical mandibles. This mandible classification distribution is quite reasonable and reveals that the M3 type, the most severe type, is relatively rare in the affected population. Additionally, in the M1 type, a milder type, patients might not notice their mild mandibular malformation or are simply unwilling to seek medical advice with no other symptoms. Thus, M3 and M1 samples are fewer in number than the other two types.

### 3.4. Correlation between OMENS+C and Disc Classifications

The correlation between OMENS+C and disc classifications among all the included samples was statistically analyzed (Table 3). A positive correlation was found between disc and mandibular defects (correlation coefficient: 0.614, *p* < 0.01), disc and ear defects (correlation coefficient: 0.242, *p* < 0.05), disc and soft tissue defects (correlation coefficient: 0.291, *p* < 0.01), and disc defects and facial clefts (correlation coefficient: 0.320, *p* < 0.01). These results indicate similar degrees of TMJ disc defects; mandibular defects; and ear, soft tissue defects, and facial clefts in HFM patients. In addition, we also found a positive correlation between orbital and soft tissue defects (correlation coefficient: 0.245, *p* < 0.05), mandible and soft tissue defects (correlation coefficient: 0.332, *p* < 0.01), mandible and facial cleft defects (correlation coefficient: 0.332, *p* < 0.01), and ear and soft tissue defects (correlation coefficient: 0.340, *p* < 0.01). These results also indicate a similar degree of defects of corresponding components in the OMENS+ system in hemifacial microsomia patients.

## 4. Discussion

When the OMENS system was first introduced, Vento et al. studied the correlation between the five components in this system and found associations between the mandible and the other four components, between orbit and soft tissue defects, between ear and nerve defects, and between nerve and soft tissue defects [15]. Additionally, there have been a few other studies about the correlation between different components in OMENS or OMENS+ systems. Wan et al. studied the correlation between the OMENS score and hearing loss in 70 HFM patients and found that the ear defect score does not correlate with hearing loss degree or hearing loss type [26]. Two other studies both found significant associations between orbital and mandibular defects and between mandibular and soft tissue defects, but they drew different conclusions when it came to other components [3,27]. There are two possible reasons for these different results. First, there is always bias between different observers when deciding the severity of symptoms, particularly for soft tissue defects, which were originally not clearly defined. CT and 3D reconstruction are always necessary to evaluate soft tissue sufficiency to fulfill surgical plans, even for experienced surgeons. The second possible reason is the special definition of facial nerve defects: N1 stands for upper-branches involvement, N2 for lower-branches involvement, and N3 for all-branches defects [1]. In other classifications, the greater the number, the more severe the symptom. For example, M3 is more severe than M2a. However, in nerve classification, N2 and N1 stand for different types of defects, with no comparison of severity, making it illogical to analyze them in the same way as other classifications. Based on these facts, more detailed classifications should be added to the OMENS+ system, and better ways to evaluate the severity of HFM symptoms are needed. In our study, the disc classification also faced problems of bias between different observers, especially when it comes to D1 and D2 to determine whether the disc length is sufficient to cover the condyle or the reconstructed condyle. Thus, quantitative evaluation in MRI images might be necessary in the refined future version. Nonetheless, a positive correlation between mandible, ear, soft tissue, cleft, and disc classifications was found in this study, and the disc classification is logical and reasonable for revealing the TMJ disc condition for hemifacial microsomia patients, and thus should be accepted as part of the OMENS+D system.

Most of the patients who visited our department were mainly seeking medical advice for malformed or asymmetrical mandibles. Treatment for HFM patients’ mandibular deformities still remains partly controversial. In the past, the degree of hemifacial microsomia deformity was used to guide treatment. Surgical treatment to reconstruct the mandible was used in children with more severe defects, while treatment for children with milder deformities could be variable [17]. Gillies first introduced the use of cartilage and bone from the rib cage to address the hypoplastic mandible in the 1920s [28]. Additionally, in 1990, McCarthy’s group began to perform MDO, and this technique became popular and widely used [29,30]. Nowadays, in order to improve facial symmetry and occlusion, CCG and MDO are the two most commonly performed operations for HFM children, though surgical procedures and surgical timing have not yet reached an agreement [16,17]. For M2b and M3 patients with unstable occlusion, MDO cannot induce the formation of the joint at the right location, and CCG has been the gold-standard technique for these types of deformities. However, soft tissue elongation, better stability, and less invasion are distinct advantages of MDO. Studies also show that the later the surgery is performed, the more stable the result could be [31]. Regarding adult hemifacial microsomia patients with milder mandibular asymmetry, combined orthognathic surgery and orthodontic treatment has been widely accepted [32]. Meanwhile, for some M2b and M3 adult patients, it is necessary to perform MDO to supply enough ramus length and soft tissue before orthognathic surgery. With the help of CT, 3D reconstruction, and 3D printing, virtual surgical planning and customized templates provide more direct and accurate surgical results [20,21]. In further studies, we plan to compare how different disc types affect surgery stability in different surgical plans. More accurate and predictive treatment plans can be expected with the help of TMJ disc classification in hemifacial microsomia patients.

When looking for possible pathogenic genes for HFM, we found many studies, but the wide spectrum of symptoms leads to numerous candidates [17,33]. The duplication of OTX2, for example, was implicated in facial cleft and mandibular hypoplasia but not orbital, ear, or nerve defects [34]. In contrast, 22q11.2 deletion caused both mandibular and ear defects in several cases [35,36] and ocular defects in other cases [37]. The haploinsufficiency of SF3B2 could cause mainly mandibular hypoplasia, and facial cleft and the absence of TMJ also appeared in some cases [38]. In addition, the mutation in TCOF1, VWA1, and 14q32 could also cause HFM symptoms [39,40]. However, regarding the animal model, only the *Hfm* heterozygous mouse model was established to study hemifacial microsomia. The phenotypes include mandibular hypoplasia and external ear abnormalities, but no detailed mapping of mutation sites or follow-up research has been conducted [41,42]. *Hoxa2* is an important gene that determines the development patterns of the first and second branchial arches of mice. *Hoxa2 ^−/−^* mice and *Hoxa2 ^flox/flox^*; *Wnt1-Cre* mice exhibit a trend of transformation from the second branchial arch to the first branchial arch and trigger rapid changes in gene expression patterns, such as *Alx4* and *Six2*; thus, abnormal expressions of *Alx4* and *Six2* could be detected in the second branchial arch at E10.5 [43,44]. Given the inspiration of animal models, more and more causative genes could be found [45,46], and it might be possible in the further study to narrow down symptoms and the number of patients, such as patients with both M3 and D3, to investigate possible causative genes for certain types of HFM.

There are three theories for the pathogenesis of HFM [33]. The first is disruption of the stapedial artery, leading to hemorrhage and subsequent deformation of the surrounding tissues; the second is abnormal migration of the subpopulation of the cranial neural crest cell migration, leading to dysmorphogenesis; and the third is the interference of Meckel’s cartilage development, which is thought to be a supplement to the first theory. According to the first theory, which was first brought up by Poswillo and verified with a rat model in 1973 [47], similar defects are expected between adjacent developing structures near the hematoma or structures with the same blood supply. This partially explains the results above, where soft tissue defects always correlate with other defects as adjacent tissues during embryogenesis. The second theory, on the other hand, suggests similar defects between structures developed from the same streams of the branchial arches. The first branchial arch gives rise to the maxilla, mandible, zygoma, trigeminal nerve, masticatory muscles, connective tissue, and a minor part of the external ear. The second branchial arch gives rise to the stapes, styloid process, portions of the hyoid bone, facial nerve, facial muscles, and the major part of the external ear. Regardless of which theory is chosen, a certain correlation should be found between the OMENS+ system components. In our study, disc classification can reveal the TMJ disc condition for hemifacial microsomia patients and help with more comprehensive treatment planning, and it should be accepted as an OMENS+D system. In addition, we found a positive correlation between the TMJ disc and mandibular defects, between the TMJ disc and ear defects, between the TMJ disc and soft tissue defects, and between the TMJ disc defect and facial cleft, which confirms the conjecture that homologous and adjacent tissues are affected to a similar degree in HFM patients.

There are several limitations to this study. First, the localization of the MRI study planes was not standardized. Due to the various structures of the TMJ in HMF patients, the long axis of condyles could not be symmetrical on both sides. To address this issue, we followed the routine of the MRI study plane: the transection plane was scanned to find the long axis of the condyle, and the sagittal plane was then determined to be perpendicular to this long axis. Additionally, for those condyles whose long axis could not be identified, the long axis of the ramus was used, and the sagittal plane was determined to be perpendicular to this long axis. Second, CCG and MDO are the two major surgical methods for HFM patients, yet there is no agreement on surgical procedures and timing. Whether this new proposed +D classification could provide support or suggestions in deciding surgical methods is still unknown. Surgical results and long-term follow-up are needed in future studies. Based on the unknown pathogenic gene and study limitations mentioned above, we have primarily planned some future research. On the one hand, we plan to narrow down the included symptoms, such as patients with both M3 and D3, to investigate possible causative genes for this most severe type of HFM using techniques such as a GWAS (Genome-Wide Association Study) or WES (Whole Exome Sequencing). Mouse models should be established for further verification and mechanism investigation. On the other hand, MDO and CCG could be performed on patients of different D classifications in order to investigate if certain surgical methods are more suitable for certain TMJ disc conditions.

## 5. Conclusions

In conclusion, this essay has proposed a new TMJ disc classification of hemifacial microsomia and has refined the diagnostic criterion as OMENS+D. With the definition of D0, D1, D2, and D3, different TMJ disc conditions could be directly revealed for HFM patients. Additionally, this D classification was proved to be positively correlated with mandibular defects, ear defects, soft tissue defects, and facial clefts, which confirms the conjecture that homologous and adjacent tissues are affected to a similar degree in HFM patients.

## Figures and Tables

**Figure 1 bioengineering-10-00595-f001:**
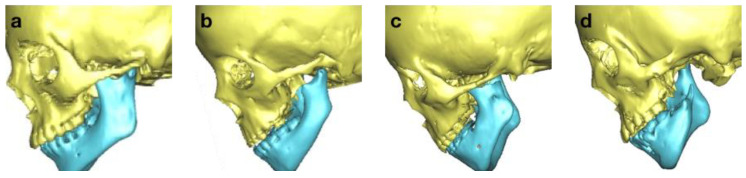
Definition of mandible classification in OMENS+ system. (**a**): small mandible and glenoid fossa with short ramus, defined as M1. (**b**): abnormally shaped and short ramus (glenoid fossa in acceptable position). (**c**): abnormally shaped and short ramus (glenoid fossa is inferiorly, medially, and anteriorly displaced with severe hypoplasia of condyle), defined as M2b; (**d**): absence of ramus and glenoid fossa, defined as M3.

**Figure 2 bioengineering-10-00595-f002:**
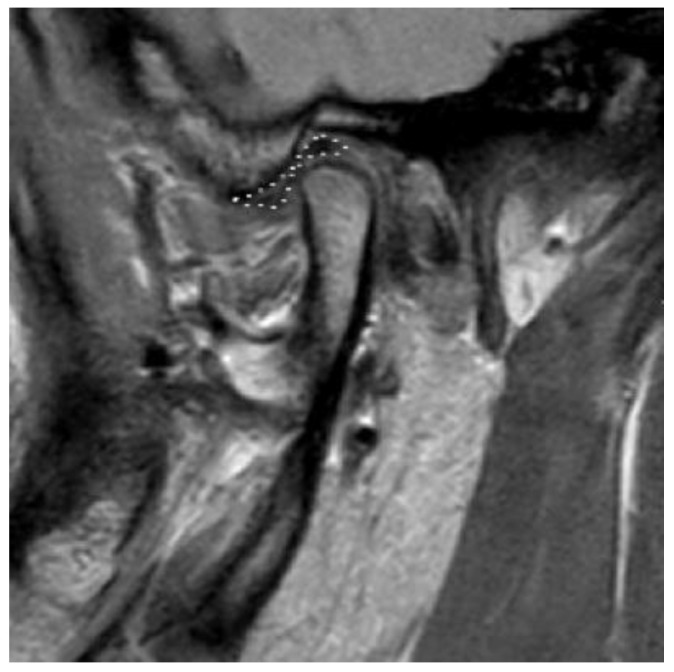
Normal disc size and shape. Defined as D0. The TMJ disc is marked in dotted white line.

**Figure 3 bioengineering-10-00595-f003:**
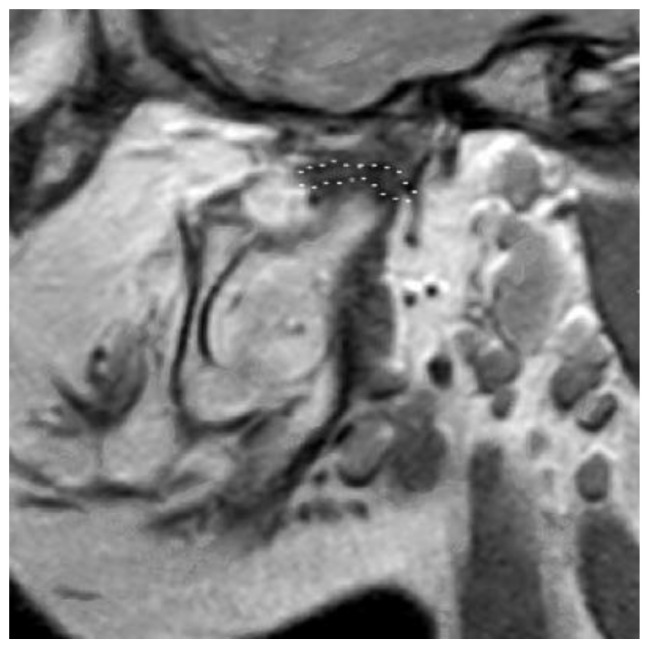
Disc malformation with adequate length to cover the (reconstructed) condyle. Defined as D1.

**Figure 4 bioengineering-10-00595-f004:**
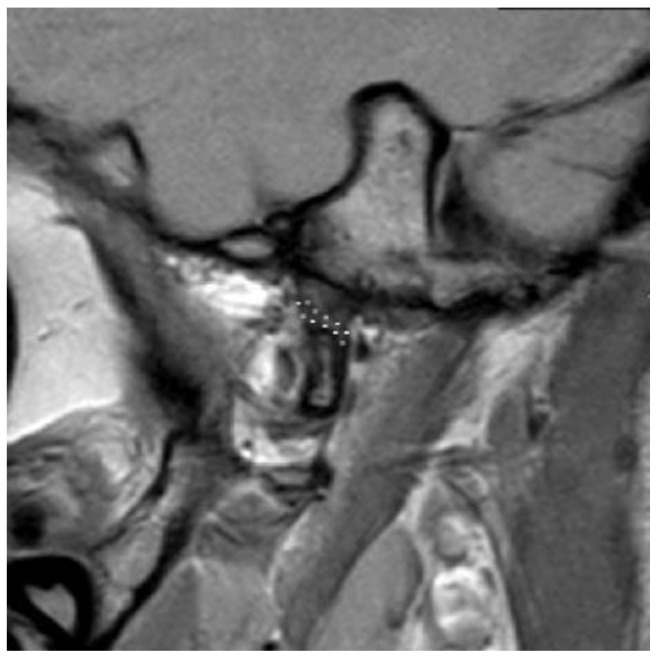
Disc malformation with inadequate length to cover the (reconstructed) condyle. Defined as D2.

**Figure 5 bioengineering-10-00595-f005:**
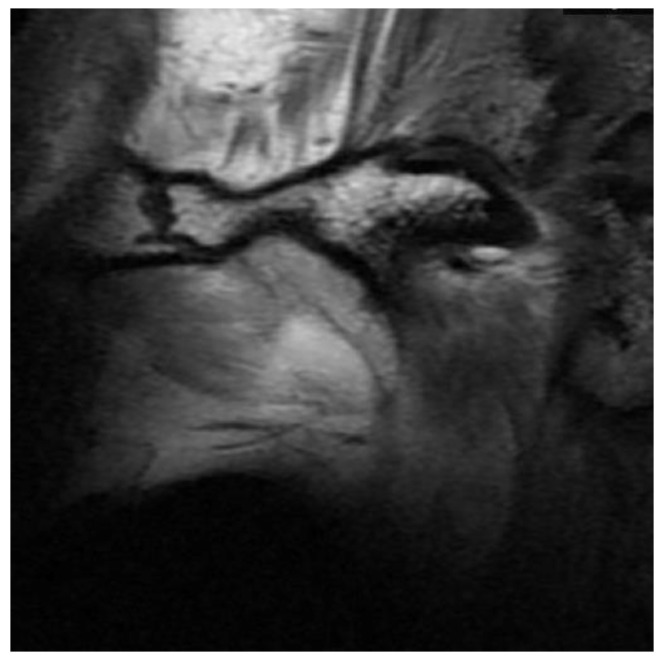
No obvious presence of disc. Defined as D3.

**Table 1 bioengineering-10-00595-t001:** Description of OMENS+ system [1,5,9,15,16,17].

O (Orbit)	
O0	normal orbital size and position
O1	abnormal orbital size
O2↓	inferior orbital displacement
O2↑	superior orbital displacement
O3	abnormal orbital size and position
M (Mandible)	
M0	normal mandible
M1	small mandible and glenoid fossa with short ramus
M2a	abnormally shaped and short ramus (glenoid fossa in acceptable position)
M2b	abnormally shaped and short ramus (glenoid fossa is inferiorly, medially, and anteriorly displaced with severe hypoplasia of condyle)
M3	absence of ramus and glenoid fossa
E (Ear)	
E0	normal auricle
E1	mild hypoplasia and cupping with presence of all structures
E2	absence of external canal with variable hypoplasia of concha
E3	malpositioned lobule with absent auricle; lobular remnant typically inferiorly and anteriorly displaced
N (Nerve)	
N0	no facial nerve involvement
N1	temporal and/or zygomatic branch involvement
N2	buccal and/or mandibular and/or cervical branch involvement
N3	all branches affected
S (Soft tissue)	
S0	no soft tissue deficiency
S1	minimal soft tissue deficiency
S2	moderate soft tissue deficiency (between S1 and S3)
S3	severe soft tissue deficiency
C (Macrostomia/Cleft)	
C0	no cleft
C1	cleft terminates medially to anterior border of masseter
C2	cleft terminates laterally to anterior border of masseter

**Table 2 bioengineering-10-00595-t002:** Distribution of samples in OMENS+C classifications and disc classification.

Orbit	** *n* **	%
O0	35	32.4%
O1	24	22.2%
O2	33	30.6%
O3	16	14.8%
Total	108	
Mandible	** *n* **	**%**
M1	15	13.9%
M2a	35	32.4%
M2b	45	41.7%
M3	13	12.0%
Total	108	
Ear	** *n* **	**%**
E0	36	33.3%
E1	12	11.1%
E2	27	25.0%
E3	33	30.6%
Total	108	
Nerve	** *n* **	**%**
N0	84	77.8%
N1	10	9.3%
N2	11	10.2%
N3	3	2.8%
Total	108	
Soft tissue	** *n* **	**%**
S0	30	27.8%
S1	41	38.0%
S2	31	28.7%
S3	6	5.6%
Total	108	
Cleft	** *n* **	**%**
C0	78	72.2%
C1	21	19.4%
C2	9	8.3%
Total	108	
Disc	** *n* **	**%**
D0	31	28.7%
D1	17	15.7%
D2	36	33.3%
D3	24	22.2%
Total	108	

**Table 3 bioengineering-10-00595-t003:** Correlation between OMENS+C and disc classification.

	Disc	Orbit	Mandible	Ear	Nerve	Soft Tissue
Orbit	−0.016					
Mandible	0.614 **	0.099				
Ear	0.242 *	0.122	0.137			
Nerve	−0.056	0.106	−0.025	0.234		
Soft tissue	0.291 **	0.245 *	0.332 **	0.340 **	0.166	
Cleft	0.320 **	−0.012	0.332 **	−0.032	0.036	0.13

* stands for *p* < 0.05, ** stands for *p* < 0.01.

## Data Availability

Not applicable.

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
