# Peer review of "A Proposal for the Classification of Temporomandibular Joint Disc Deformity in Hemifacial Microsomia"

_bioengineering, 2023, doi:10.3390/bioengineering10050595_

Round 1

Reviewer 1 Report (Previous Reviewer 1)

Thank you for resubmitting your manuscript.

Thank you for resubmitting your manuscript.

Author Response

Thank you for your advice.

English editing was finished.

Reviewer 2 Report (Previous Reviewer 3)

The Authors have adequately addressed the Reviewers' comments. Further improvement in style, spelling and grammar is needed.

An interesting contribution to the literature and our understanding of HFM.

Further improvements are needed in style, grammar and spelling.

Author Response

Thank you for your advice.

English editing was finished, and the style, grammar and spelling are improved.

This manuscript is a resubmission of an earlier submission. The following is a list of the peer review reports and author responses from that submission.

Round 1

Reviewer 1 Report

I read the manuscript with interest.

We need to realize that there is important information missing such as a section for the limitations of the study, future research, and conclusions.

Thus, the. authors should work on this to fulfill the requirements for submission.

Reviewer 2 Report

Dear authors, despite the efforts, the manuscript does not provide sufficient evidence to support your proposed new classification. 

1) "3.2. Definition of disc classification": this paragraph is arbitrary. Authors should explain what makes this classification evidenced based.

2) "3.4. Correlation between OMENS+C and disc classifications-Based on the above results, disc classification is able to reveal the TMJ disc deformity  for hemifacial microsomia patients. It is possible to use this classification to evaluate TMJ condition, thus to help with more comprehensive treatment planning and prognosis of stability of surgical orthodontic treatment. In conclusion, it should be accepted as OMENS+D system.": first of all, this paragraph should not be in the discussion section, but in the conclusion section supported by sufficient evidence which frankly I fail to see. Why the correlation between the disk and the existent classification should provide evidence to switch to a new classification?

3)The discussion section does not discuss your proposed classification but the existing evidence on the topic. 

In conclusion, I think this paper is not suitable for pubblication in the present form. If the authors are able to rearrenge it and focus it on providing scientific evidence on their proposed new classification, they should resubmit it. 

Kind regards

Reviewer 3 Report

The Authors have studied the TMJ Disc in patients with HFM and presented an enhancement of the OMENS classification system. They are to be commended for addressing this neglected area of interest. 

Did any of the subjects present signs or symptoms of TMD?

What steps were taken to standardize the localization of the MRI study planes given the distorted anatomy of the region?

The Authors chose a classification system based on whether the disc would be adequate for reconstructive surgery. Would another system based on anatomy and deviation from the normal be more appropriate? I.E position of the posterior band vis a vis condyle, relationship of mid zone to Condyle/eminence anatomy, thickness of disc, etc. The other parts of the OMENS system compare the distorted anatomy to the syndromic anatomy-would this be a better, more consistent approach? The importance of the Disc classification to surgical repair can be handled separately. 

The paragraph on genetic mechanisms is interesting but not of primary concern-can it be eliminated or reduced? 

The paragraph on mechanisms is of greater interest-can it be expanded-do animal models throw any light on the subject. Has anyone studied this? How does the notion of the TMJ developing from three Blastemas fit with these results?

Thank you for looking into this area. 

Reviewer 4 Report

Novel and clinically relevant/sound suggestion. Validity should be addressed in a bigger cohort and externally in subsequent publications